# FDDS: Feature Disentangling and Domain Shifting for Domain Adaptation

**Huan Chen [1], Farong Gao [1,2,*] and Qizhong Zhang [1,2]**

1   HDU-ITMO Joint Institute, Hangzhou Dianzi University, Hangzhou 310018, China;
    202320035@hdu.edu.cn (H.C.); zqz@hdu.edu.cn (Q.Z.)
2   School of Automation, Hangzhou Dianzi University, Hangzhou 310018, China
*   Correspondence: frgao@hdu.edu.cn

**Abstract:** Domain adaptation is a learning strategy that aims to improve the performance of models in the current field by leveraging similar domain information. In order to analyze the effects of feature disentangling on domain adaptation and evaluate a model's suitability in the original scene, we present a method called feature disentangling and domain shifting (FDDS) for domain adaptation. FDDS utilizes sample information from both the source and target domains, employing a non-linear disentangling approach and incorporating learnable weights to dynamically separate content and style features. Additionally, we introduce a lightweight component known as the domain shifter into the network architecture. This component allows for classification performance to be maintained in both the source and target domains while consuming moderate overhead. The domain shifter uses the attention mechanism to enhance the ability to extract network features. Extensive experiments demonstrated that FDDS can effectively disentangle features with clear feature separation boundaries while maintaining the classification ability of the model in the source domain. Under the same conditions, we evaluated FDDS and advanced algorithms on digital and road scene datasets. In the 19 classification tasks for road scenes, FDDS outperformed the competition in 11 categories, particularly showcasing a remarkable 2.7% enhancement in the accuracy of the bicycle label. These comparative results highlight the advantages of FDDS in achieving high accuracy in the target domain.

**Keywords:** feature disentangling; domain adaptation; attention mechanism; adaptation separation; domain shifting

**MSC:** 68T05

## 1. Introduction

Compared with traditional machine learning methods, deep learning has powerful feature-extraction and feature-processing capabilities for solving big data problems and has achieved remarkable results [1,2]. Nevertheless, when neural networks attempt to generalize across domains, domain shifting will lead to decreases in performance. It is challenging to solve this problem [3]. Researchers have made extensive efforts in domain adaptation, exploring various approaches to facilitate cross-domain generalization [4,5]. Among these strategies, the feature domain adaptation and pixel-level domain adaptation methods are two of the most prominent.

Feature domain adaptation methods [6,7] learn domain-invariant features by applying a feature-extraction model in a feature space [8] or a generative adversarial network model [9]. However, the training process of these methods is difficult to visualize, and there may be difficulties in capturing domain shifting at the pixel level and some edge information. In addition, pixel-level domain adaptation methods [10,11] utilize generative adversarial networks (GANs) [12] to adjust the source domain image in the original pixel



space, making it resemble an image drawn from the target domain. Both feature- and pixel-level domain adaptation methods have advantages, and they operate in different ways [13]. As a result, some methods [14,15] combine feature domain adaptation with pixel-level domain adaptation to enhance the performance of domain adaptation from different aspects and achieve significant results.

In recent years, some methods [16,17] have attempted to separate image features into a potential feature space and extract shared and private components from source and target domain image samples. The results have shown that feature disentangling has a positive effect on extracting domain-shared features to a certain extent, thereby improving the classification accuracy of the target domain's model [16]. However, these methods often focus on the shared features of the two domains while overlooking the private features that may limit performance improvement to some degree [15]. In this context, one method [18] introduced a state-of-the-art separation methodology that linearly divided features into content and style components, albeit with a fixed 1:1 ratio for disentangling content and style features. This fixed ratio may reduce the adaptability of models across diverse scenes. Oppositely, another model proposed in method [15] incorporated a non-linear disentangling manifold allowing for flexible determination of the proportion of content and style features during model training. Therefore, it is crucial to study feature disentanglement comprehensively.

In a specific scenario, one model needs to fulfill the requirements of the target domain while maintaining a relatively accurate recognition ability in the source domain. For instance, when a network model operates on a server, it should use its functions from the source domain while serving clients in the target domain. To tackle this challenge, a traditional approach is to train the network model exclusively for the target domain and then transfer it back to the source domain when necessary. However, this iterative process results in a significant waste of training resources and fails to meet the needs of both domains adequately. It remains challenging for a compressed or service-providing model to simultaneously address the services of both domains [14].

To address the aforementioned issues, we propose a Feature Disentangling and Domain Shifting (FDDS) method for domain adaptation. This approach utilizes a non-linear disentangling technique to separate features from two domains into content and style components. The proportion of content features in different domains is determined by learnable weights, allowing for more accurate feature separation based on the specified proportion. Furthermore, in addition to providing additional information for feature disentangling, our method incorporates a domain shifter that enhances the performance of the target domain while ensuring the performance of the source domain to the maximum extent possible. Unlike the data calibrator proposed in previous methods [14], our domain shifter incorporates a dual-attention mechanism, involving both spatial and channel attention [19]. This mechanism enables the model to focus on key points within the scene, capture more valuable image information, and disregard irrelevant details, thereby improving the efficiency of image-processing.

The contributions of our approach are summarized as follows:

1.  We proposed a new non-linear feature disentangling method, which determines the proportion of content features and style features in source and target domains through learnable weights. This approach enables the precise separation of content and style features based on their corresponding proportions;
2.  We integrated a dual-attention mechanism involving both spatial and channel attention into the domain shifter network, which preserves the performance of the source domain after domain shifting. As a result, our model can seamlessly transition between serving the source and target domains;
3.  To evaluate our approach, we conducted experiments in the digit classification and semantic segmentation tasks. Our method exhibited superior performance compared to competing approaches, particularly in the semantic segmentation tasks. Specifically,

our FDDS method outperformed the competition in 11 out of 19 classification labels and achieves optimal or suboptimal results in 16 categories.

This paper is structured as follows. In Section 1, we introduce the background of the research, which led to our work. And in Section 2, we introduce the related work in the field. Section 3 presents the method details of FDDS. Moving on to Sections 4 and 5, we demonstrate and analyze the numerical results of FDDS on public datasets and discussed ablation experiments. Finally, we conclude by summarizing the text in full.

## 2. Related Work

### 2.1. Deep Domain Adaptation

Transfer learning is a methodology that utilizes known information to learn new knowledge in unknown fields. Domain adaptation is a type of transfer learning and serves as a prevalent technique for addressing the transferability of diverse datasets. After Yosinski et al. [20] explored how to transfer features in neural networks, many feature domain adaptation methods have emerged. Tzeng et al. [21] built upon this by incorporating an adaptation layer into the AlexNet, proposing the deep domain confusion (DDC) method. Long et al. [22] expanded upon DDC, arguing that multi-layer adaptation was superior to single-layer adaptation, and created the deep adaptation network (DAN). Sun et al. [23] used the feature scale factor to express the relative importance of features, and captured the internal manifold structure of data in the low-dimensional manifold subspace, thus reducing the probability distribution between different domains. Recently, GANs [12] has become more and more popular. Ganin et al. [24] first combined domain adaptation with GANs, proposing the domain-adversary neural network (DANN) which leverages GANs to improve the network's feature extraction capability. Akada et al. [25] used GANs to learn the domain invariant features of the network by self-supervised learning, and complete the transfer from synthetic images to real images.

In order to enhance the interpretability of neural networks, scholars have conducted a series of studies on pixel-level domain adaptation. Bousmalis et al. [26] utilized a generative adversarial network to align the distribution of the source domain with the target domain in pixel space. Pixel-level domain adaptation is beneficial for both assigning labels to images and improving feature-level domain adaptation. Hence, researchers have attempted to integrate these two methods. Hoffman et al. [13] were the first to incorporate feature and pixel-level domain adaptation, introducing cycle-consistent loss to augment the model's semantic consistency. Ye et al. [14] further improved the network's accuracy in the target domain by introducing a data calibrator, while at the same time ensuring that the classifier retains its ability to accurately classify the source domain. In this paper, we sought to improve upon the data calibration mechanism by embedding an attention mechanism into the network. The attention mechanism [27] enables the network to focus on key features in the scene, thereby further enhancing the model's classification accuracy in the source domain.

### 2.2. Attention Mechanism in Image Generation

The attention mechanism (AM) was initially applied in the domain of machine translation. With the development of AM in recent years, AM has emerged as an important solution for addressing the issue of information overload in the field of image generation. Chen et al. [28] proposed attention-GAN, which leverages the AM to transform the specific position of an image while leaving the background unaffected, thus demonstrating the feasibility of cooperative functioning between the AM and the GAN. Emami et al. [29] introduced self-attention generative adversarial networks (SAGAN) that capture long-distance dependencies via the AM, enabling the generated image to represent global features, resulting in significant success in the field of image generation. Daras et al. [30] proposed the design of a two-dimensional local attention mechanism to generate the model. By reducing the attention feature map, operation efficiency was accelerated, and the model became lightweight. Woo et al. [31] proposed CBAM, which combined the channel and spatial

attention mechanisms on features, and achieved better results than the single attention mechanism. Following this, many methods [32] incorporated the attention mechanism into the network as a component to improve image generation performance. In this paper, we incorporate the AM into the domain shifter network, effectively rendering it lightweight and better-equipped to extract key features from the scene.

*2.3. Feature Disentangling*

In the field of domain adaptation, the research on feature separation methods has recently aroused people's interest. Bousmalis et al. [16] learned to divided features into two components: private and shared features, and demonstrated that the modeling of private features is helpful to extract domain-invariant features. Gonzalez-Garcia et al. [33] attempted to separate the private factors in both fields from those that were shared across fields. Liu et al. [17] proposed a cross-domain feature disentangling, which can connect information and transmit attributes across data domains. Zou et al. [34] proposed a joint learning framework to separate identity-related/irrelevant features for personnel re-identification tasks.

Feature disentangling can also decompose features into style features and content features. This disentangling method was initially employed in the domain of style transfer and was extensively investigated in the context of artistic styles. Tenebaum et al. [35] demonstrated how perceptual systems separate content and style and proposed a bilinear model to address these two factors. Elgammal et al. [36] introduced a method for separating style and content on the manifold that represents dynamic objects. Gatys et al. [37] presented a way to manipulate the content and style of natural images by leveraging the universal feature representation of convolutional neural networks (CNN) learning. Zhang et al. [18] linearly separated features into content components and style components. However, Lee et al. [15] argued that features are not necessarily linearly separable in real-life scenarios and proposed a non-linear disentangling method to isolate potential variables on non-linear manifolds. In this paper, we also employ a non-linear disentangling method to determine the proportion of content features in different fields using learnable weights, leading to more accurate feature separation based on the content proportion.

## 3. Materials and Methods

*3.1. Model Description*

The network architecture of our method is shown in Figure 1. The FDDS network consists of a domain shifter $DS$, an encoder $E$, a feature separator $S$, a generator $G$, two discriminators of source and target domains $D_A, D_B$ and a perceptual network $P$.

In the FDDS network, source and target images $I_A, I_B$ are fed into the model and initially processed by the domain shifter $DS$, which is processed $I_A$, $I_B$ into $I_{DSA}$, $I_{DSB}$ and captures and stores critical information necessary to preserve the model's classification ability for the source domain. The images are then passed through the encoder $E$ to extract their features $F_A, F_B, F_{DSA}, F_{DSB}$, respectively. The feature separator utilizes feature merging factors $\lambda_1, \lambda_2$ to merge features and decompose the merged features $F_A{}', F_B{}'$ into content and style components, where content features are represented by $C_A, C_B$ and style features are represented by $S_A, S_B$. The generator $G$ maps the original features $F_A, F_B$ and transfers features $F_{A \to B}, F_{B \to A}$ into the image space. In addition, a pre-trained perceptual network $P$ is employed to extract perceptual features and enforce constraints on content and style similarities. In the training phase, two discriminators $D_A, D_B$ are utilized to impose adversarial loss constraints on both source and target domains.

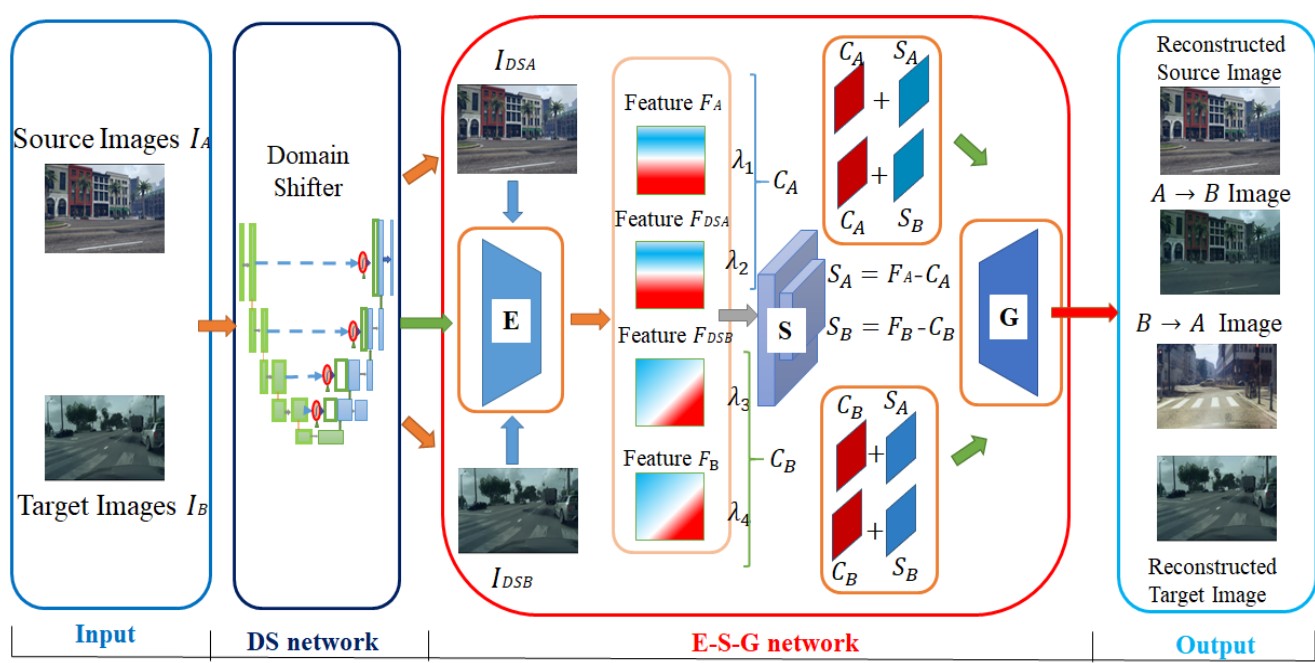

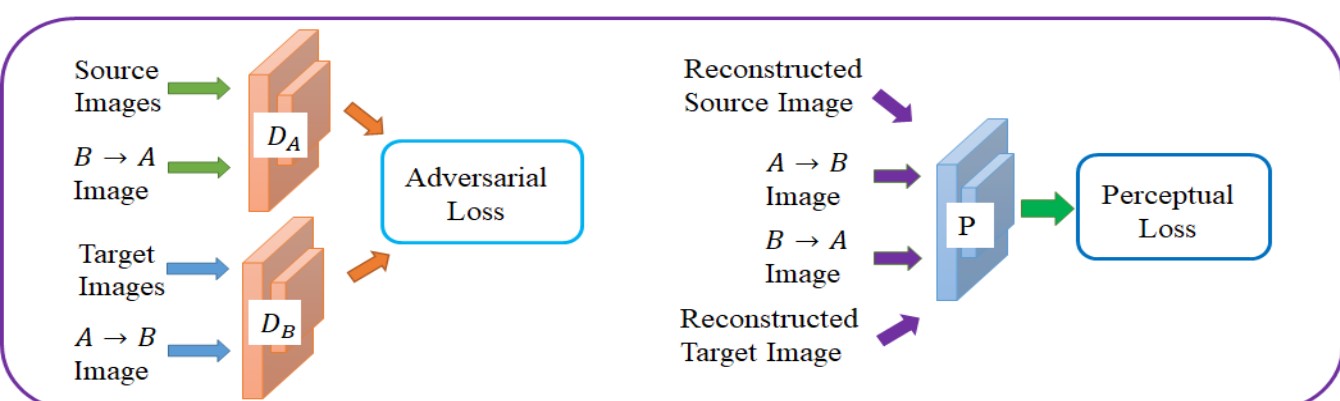

**Figure 1.** The network architecture of FDDS: (**up**) The network is composed of a domain shifter, an encoder, a feature separator, a generator, and the input and output of the model; and (**down**) the adversarial loss and perceptual loss of the network.

### 3.2. Domain Shifter Network

The first step of our method involves utilizing the domain shifter to process the input image. The network structure of domain shifter is composed of U-Net [38], with the dual attention mechanism of spatial and channel added during each up-sampling to capture the inter-channel dependencies and intra-pixel spatial relationships, respectively. This design enables more effective feature extraction and ultimately leads to superior performance results [19]. The network architecture of the domain shifter is shown in Figure 2.

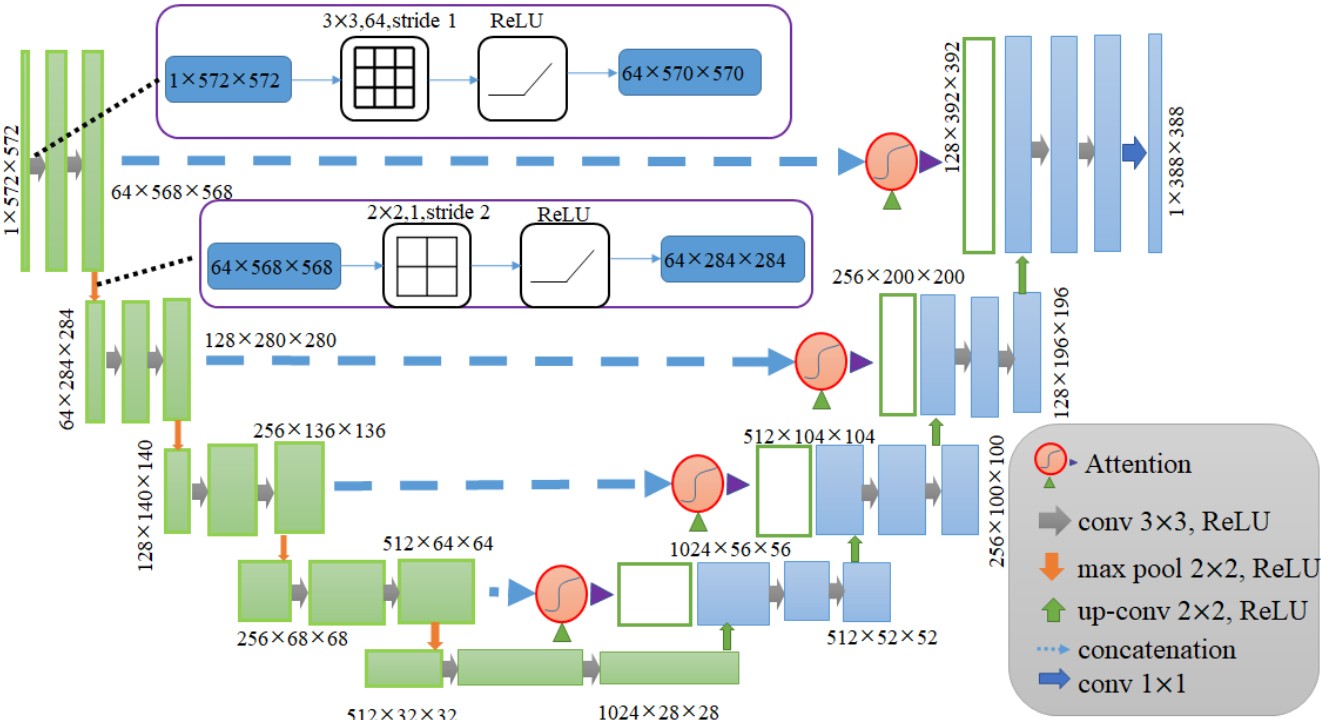

**Figure 2.** Domain shifter network architecture. (The two purple boxes in the figure represent two network structures in first layer, and the number of channels and step sizes of other modules not shown may be slightly different, which can be inferred by using the feature numbers of different dimensions of the results).

As shown in Figure 2, our domain shifter adopts the U-Net with an attention mechanism and two features to extract the network and uses ReLU as the activation function. The right three and the corresponding feature map are shown on the left. The features on the left require attention-mechanism processing and fusion.

The structure of the attention mechanism is shown in Figure 3.

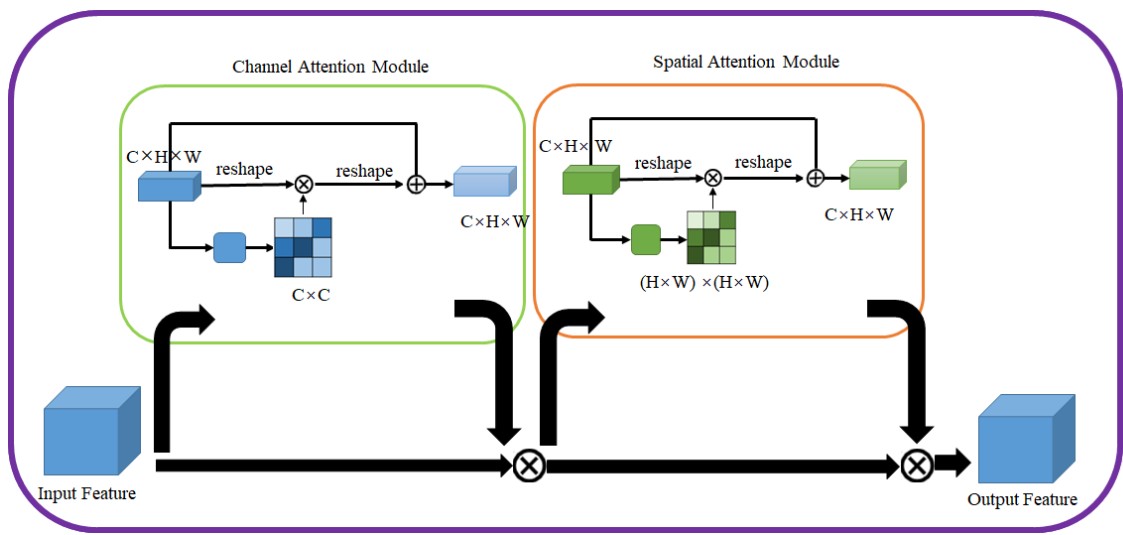

**Figure 3.** The structure of the attention mechanism ($\otimes$ represents matrix multiplication element by element, while $\oplus$ represents feature fusion).

As shown in Figure 3, we adopted a dual-attention mechanism with channel and spatial concatenation to complete feature extraction and fusion. In Figure 3, the solution of the channel and spatial attention mechanism is as follows.

$$
\begin{aligned}
F' &= M_c(F_{in}) \otimes F_{in}, \\
F'' &= M_s(F') \otimes F', \\
F_{out} &= F' \oplus F''
\end{aligned}
\tag{1}
$$

where $F_{in}$ is the input feature, $F'$ is the feature after channel attention processing, $F''$ is the image after spatial attention processing, and $F_{out}$ is the output feature; $\otimes$ represents the matrix multiplication element by element, and $\oplus$ represents feature fusion.

The primary aim of introducing the domain shifter into the network is to ensure the accuracy of the target domain while preserving the classification accuracy of the source domain. Specifically, the domain shifter tries to produce analogous results for both the processed target domain image and the unprocessed source domain image when fed into the final classifier. This approach effectively preserves the classifier's classification ability for the source domain, thereby ensuring its accuracy. To achieve this outcome, the domain shift must satisfy certain conditions, including but not limited to,

$$
Y(I_{DSB}) \sim Y(I_A), Y(I_{DSA}) \sim Y(I_A)
\tag{2}
$$

where $Y(I_A)$ represents the classification result obtained by the final classifier when the input $I_A$, and the same is true for others.

In order to make the classification result better, the domain shifter's constraints are as follows,

$$
I_{DSA} \sim I_A, I_{DSB} \sim I_A
\tag{3}
$$

The domain shifter deceives the discriminator $D_d$ in the model by the following loss function,

$$
\mathcal{L}_{DS}^d = \mathbb{E}[y_a \log(D(x_a)) + y_b \log(D(x_b))]
\tag{4}
$$

where $x_a, x_b$ represent the sample of the source and the target domain, respectively, and $y_a, y_b$ represent the label of the source and the target domain, respectively. $\mathbb{E}$ represents mathematical expectation.

The domain shifter component in the network is trained to adhere to Equation (3). During the training process, the domain shifter captures and stores specific perturbations between source and target domain images, while disregarding other irrelevant information. When training with the source domain, the loss brought by the domain shifter is minimal, resulting in an almost identical mapping. Conversely, when training with the target domain, the domain shifter is required to facilitate a transformation that makes the target domain image more closely resemble the source domain image to deceive the discriminator and achieve optimal performance.

$$
I_{DSB} = I_B + j
\tag{5}
$$

where $j$ represents the perturbation learned by the domain shifter in the target domain.

### 3.3. Feature Disentangling Module

The generalization ability of deep adaptation domain network model largely depends on the quality of feature disentangling. Therefore, we enhanced the functionality of the feature disentangling module.

Subsequent to the domain shifter processing the image, the input images $I_A, I_B$ of both the source and target domains, along with the image $I_{DSA}, I_{DSB}$ that has been processed

by the domain shifter, are fed into the encoder $E$ for encoding. This conversion process translates image information into feature information,

$$F_A = E(I_A), F_{DSA} = E(I_{DSA})$$
$$F_B = E(I_B), F_{DSB} = E(I_{DSB})$$

(6)

where $E(I_A)$ represents features of encoding the image $I_A$ with the encoder $E$, and the same is true for others. $F_A, F_B$ represented the features of source and target domains, respectively.

Subsequent to obtaining the features $F_A, F_B, F_{DSA}, F_{DSB}$ through the encoder, the feature separator $S$ will partition these features into two distinct categories, namely content features $C_A, C_B$ and style features $S_A, S_B$. Content features $C_A, C_B$ are achieved via processing features by $S$ and multiplying $\omega_A, \omega_B$ separately. In contrast, style features $S_A, S_B$ are generated by subtracting content features $C_A, C_B$ from two-domain features $F_A, F_B$. This non-linear method of disentangling serves to completely separate content and style features. In separator $S$, non-linear mapping is also used to ensure the accuracy of content features [36]. As a result, Equation (7) accurately represents the methods used for processing image samples in both the source and target domains,

$$C_A = \omega_A S(\lambda_1 F_A + \lambda_2 F_{DSA}), S_A = F_A - C_A, \text{where } \lambda_1 + \lambda_2 = 1.$$
$$C_B = \omega_B S(\lambda_3 F_B + \lambda_4 F_{DSB}), S_B = F_B - C_B, \text{where } \lambda_3 + \lambda_4 = 1.$$

(7)

where $\omega_A, \omega_B$ represent the weight parameters for the distribution of the source and target domains in the content space that has been standardized. The purpose of these parameters is to minimize feature deviation. $\lambda_1, \lambda_2$ are balance coefficients for source features $F_A$ and features $F_{DSA}$, respectively. These coefficients play a critical role in proportionately integrating features $F_A, F_{DSA}$. Meanwhile, $\lambda_3, \lambda_4$ serve as balance coefficients for the target features $F_B$ and features $F_{DSB}$. During training, the value of these coefficients is determined. The content feature is attained by applying a non-linear function and the learnable feature-scale parameter $\omega_A, \omega_B$. On the other hand, the style feature is calculated by subtracting content components from the entire feature.

Feature disentangling is used to transfer features across domains [18], and the specific combination method is as follows,

$$F_{A \to B} = \omega_{A \to B} C_A + S_B, F_{B \to A} = \omega_{B \to A} C_B + S_A,$$
$$where \; \omega_{A \to B} = \frac{\omega_B}{\omega_A}, \omega_{B \to A} = \frac{\omega_A}{\omega_B}$$

(8)

where $\omega_{A \to B}, \omega_{B \to A}$ are the weight parameters for the distribution of domain-transfer images $I_{A \to B}, I_{B \to A}$ in the standardized content space, and the transfer domain features $F_{A \to B}, F_{B \to A}$ are synthesized using learnable scale parameters $\omega_{A \to B}, \omega_{B \to A}$.

During the processing of FDDS, the model can learn all the parameters of Equation (8). Upon undergoing $S$ processing, content features $C_A, C_B$ and style features $S_A, S_B$ are recombined to produce novel domain transfer features $F_{A \to B}, F_{B \to A}$. Then, these transferred features $F_{A \to B}, F_{B \to A}$, along with the features $F_A, F_B$ in Equation (6) are projected into image space through the generator $G$. This leads to the creation of new images $I_{A \to B}, I_{B \to A}, I'_A, I'_B$ that offer greater value when utilized with a loss function [15]. The image generation method is as follows,

$$I_{A \to B} = G(F_{A \to B}), I_{B \to A} = G(F_{B \to A}),$$
$$I'_A = G(F_A), I'_B = G(F_B),$$

(9)

where $I_{A \to B}, I_{B \to A}$ is the domain adaptation image and $I'_A, I'_B$ is the reconstructed image.

*3.4. Training Loss*

FDDS uses domain shifter $DS$, encoders $E$, feature separators $S$ and generators $G$ to train the network by minimizing the overall network's loss function $\mathcal{L}^d$, while the discriminator $D_d$ tries to maximize it,

$$\min_{DS,E,S,G} \left( \sum_{d \in \{A,B\}} \max_{D_d} \mathcal{L}^d \right) \tag{10}$$

where the domain $d$ is the source domain $A$ or the target domain $B$.

The overall loss of the model includes reconstruction loss $\mathcal{L}_{Rec}$ with balance factor $\alpha_i$, consistency loss $\mathcal{L}_{Con}$, perceptual loss $\mathcal{L}_{Per}$ and adversarial loss $\mathcal{L}_{GAN}$ and domain shifter loss $\mathcal{L}_{DS}^d$,

$$\mathcal{L}^d = \alpha_1 \mathcal{L}_{Rec}^d + \alpha_2 \mathcal{L}_{GAN}^d + \alpha_3 \mathcal{L}_{Con}^d + \alpha_4 \mathcal{L}_{Per}^d + \alpha_5 \mathcal{L}_{DS}^d \tag{11}$$

where $\mathcal{L}_{DS}^d$ has been discussed in Equation (4), and the following are the details of the remaining losses.

(a) Reconstruction loss: Loss $\mathcal{L}_R$ is used to represent that the difference between the input image $I_d$ and the reconstructed image $I_d'$ is minimized after $E$ and $G$ training,

$$\mathcal{L}_{Rec}^d = \mathcal{L}_R(I_d, I_d'), \text{where } I_d' = \underset{i \in \{1,3\}, j \in \{2,4\}}{G} (E(\lambda_i I_d + \lambda_j I_{DSd})) \tag{12}$$

(b) Adversarial loss: Two discriminators $D_{d \in \{A,B\}}$ are used to evaluate the countermeasure loss on the source and the target domain, respectively [13]. The following is the countermeasure loss of source domain $A$ to target domain adaptation $B$,

$$\mathcal{L}_{GAN}^B = \mathbb{E}_{x_b \sim P_{data(X_b)}} [\log D_B(x_b)] \\ + \mathbb{E}_{(x_a, y_a) \sim P_{data(X_a, Y_a)}} [\log(1 - D_B(I_{A \to B}(x_a, y_a)))] \tag{13}$$

where $x_b$ represents the sample of the target domain, and $(x_a, y_a)$ represents the sample and label of the source domain.

For the adaptation from the target domain $B$ to the source domain $A$, the same adversarial loss is also imposed. And the standardization is applied to all layers in $G$ and $D$, and the discriminator is used for complex scenes, such as road-scene adaptation, together with adversarial loss.

(c) Consistency loss: Consistency loss [13] attempts to preserve content and style modules after re-projecting the domain-transfer image into a representation space represented by,

$$\mathcal{L}_{Con}^A = \mathcal{L}_R(C_A, C_{A \to B}) + \mathcal{L}_R(S_A, S_{B \to A}) \tag{14}$$

where $\mathcal{L}_R$ means to minimize the difference between the input image $I_d$ and the reconstructed image $I_d'$ after training in $E$ and $G$, and the content factor $C_{A \to B}$ and style factor $S_{B \to A}$ are extracted from the domain-transfer image $I_{A \to B}, I_{B \to A}$ through the same encoder $E$ and separator $S$, respectively. This loss function serves as a clear incentive to maintain consistency in scene structure during the process of domain adaptation.

(d) Perceptual loss: In traditional semi-supervised training, the class labels are utilized as semantic indicators that guide feature disentangling. Conversely, frame training is capable of disentangling features without the requirement of labeled data. To facilitate the unsupervised learning of feature disentanglement, we implemented a perceptual loss [39] in the network, which is a widely employed framework in style transfer. This is defined as follows,

$$\mathcal{L}_{Per}^A = \mathcal{L}_{Content}^A + \lambda \mathcal{L}_{Style}^A \\ \mathcal{L}_{Per}^B = \mathcal{L}_{Content}^B + \lambda \mathcal{L}_{Style}^B \tag{15}$$

where $\mathcal{L}_{Content}^A$, $\mathcal{L}_{Content}^B$ are content loss and style loss $\mathcal{L}_{Style}^A$, $\mathcal{L}_{Style}^B$ [15]. Defined as follows,

$$
\begin{aligned}
\mathcal{L}_{Content}^B &= \sum_{l \in L_c} \|P_l(I_A) - P_l(I_{A \to B})\|^2 \\
\mathcal{L}_{Style}^B &= \sum_{l \in L_S} \|G_1(P_l(I_B)) - G_1(P_l(I_{A \to B}))\|_F^2
\end{aligned}
\tag{16}
$$

where the set of layers $L_c$, $L_s$ is a subset of the perceptual network $P$. The weight parameter $\lambda$ balances two losses, and $G_1$ is a function of the matrix, given each layer's feature $l$. We also applied batch normalization to better-stylize the process.

## 4. Results

### 4.1. Digit Classification

4.1.1. Dataset

The MNIST dataset [40] is widely employed for handwritten digit recognition, including 60,000 training images and 10,000 test images. The SVHN dataset [41] consists of a vast collection of house number images that have been extracted from Google Street View. Given that the images in SVHN are real-world images of house numbers taken from street-level, they present greater difficulty with varying styles and backgrounds. With 73,257 training images and 26,032 testing images, SVHN is a sizeable dataset. Yet another typical handwritten digit recognition dataset is the USPS dataset [42], featuring over 20,000 pictures.

4.1.2. Baselines and Implementation Details

- Source Only: The classifier trained in the source domain is directly used in the target domain;
- DANN [24]: GAN is used to improve the feature extraction ability of the network;
- DSN [16]: Decouple features from private features and common features, and identify the target domain through common features;
- ADDA [9]: GAN method based on discriminative model;
- CyCADA [13]: Combine features from features and pixels, and introduce cyclic loss into domain adaptive learning;
- GTA [43]: Using the ideas of generation and discrimination, learning similar features by using GAN;
- LC [14]: It is equally important to put forward the lightweight calibrator component and start to pay attention to the performance of the source domain;
- DRANet [15]: Decouple features into style and content features, and propose nonlinear decoupling;
- CDA [44]: Using two-stage comparison to learn good feature separation boundary;
- Target Only: Training directly in the target domain and testing in the target domain is equivalent to supervised learning.

In the digit classification task, our initial network adopted LeNet; used the training set of the source domain to train the network under supervision; and used the task loss to make it a classifier of the source domain. In order to evaluate the above model, we used the source code provided by the author and some experimental data provided by LC [14]. The FDDS model was implemented in Pytorch. All the models in this paper were trained on a single NVIDIA GTX 2080 GPU using CUDA11.7. The running time of our method is about 4~6 h.

FDDS sets hyper-parameters as follows. In order to train the source domain classifier, we added adversarial loss and perceptual loss to the network. During the source domain classifier's training, we set the learning rate to $1 \times 10^{-4}$ and the batch size to 128, then trained 200 epochs to add to the network. In the process of domain shifting, we refer to the training parameter setting of LC [14], and set the learning rate to $1 \times 10^{-5}$, the batch size to 128, and trained 200 epochs. The reason why the learning rate decreases in the domain shifting process is that we do not want to update the generator parameters too early, which would result in a poor training effect.

4.1.3. Experimental Process

We evaluated the performance of the FDDS model on three prevalent digit datasets: MNIST [40], SVHN [41], and USPS [42]. The network training used the identical data-processing and LeNet architecture as [13], and performed three unsupervised domain adaptation tasks: MNIST to USPS, USPS to MNIST and SVHN to MNIST.

We set up two experimental groups: MNIST to USPS and USPS to MNIST. We wanted to train a classification model that performed well on both USPS and MNIST tasks. In this context, MNIST and USPS are each other's source and target domains, and the experiments aimed to investigate the effect of domain exchange between the two domains.

In the SVHN to MNIST task, while the SVHN samples differed significantly in background and scale from those of the MNIST, the digital shape of the primary content remained relatively unchanged. In contrast, the digital shape of handwritten digits in the MNIST was subject to significant variability due to handwriting, thereby presenting a well-defined yet challenging domain adaptation scene. Lastly, the model was assessed using 1000 MNIST samples to gauge its performance.

Therefore, we conducted three groups of experiments, MNIST to USPS, USPS to MNIST and SVHN to MNIST, and compared FDDS with other competitive methods, in which the source only and target only were used as the control group, which, respectively, represented the results of training only using the source/target domain; DANN [24], DSN [16], ADDA [9] and GTA [43] use traditional feature separation methods to separate features into private and shared parts; CyCADA [13] and DRANet [15] separate features into style and content; and LC [14] and CDA [44] both use lightweight components and confrontation generation networks to realize domain adaptation. The numerical results of these methods are shown in Table 1.

**Table 1.** Accuracy comparison of FDDS to state-of-the-art methods on domain adaptation for digit classification (%).

| Method | MNIST to USPS | USPS to MNIST | SVHN to MNIST |
|---|---|---|---|
| Source Only | 80.2 | 44.9 | 67.1 |
| DANN (2014) [24] | 85.1 | 73.0 | 70.7 |
| DSN (2016) [16] | 85.1 | - | 82.7 |
| ADDA (2017) [9] | 90.1 | 95.2 | 80.1 |
| CyCADA (2018) [13] | 95.6 | 96.5 | 90.4 |
| GTA (2018) [43] | 93.4 | 91.9 | 93.5 |
| LC (2020) [14] | 95.6 | 97.1 | **97.1** |
| DRANet (2021) [15] | <u>97.6</u> | 96.9 | - |
| CDA (2023) [44] | 96.6 | <u>97.4</u> | 96.8 |
| **Ours** | **98.1** | **97.6** | <u>96.9</u> |
| Target Only | 97.8 | 99.1 | 99.5 |

The optimal performance is bold, and the suboptimal performance is underlined.

As shown in Table 1, FDDS, as an unsupervised learning algorithm, achieved the same performance as directly applying labeled target domain training in the MNIST to USPS task. This is due to the network structure's ability to increase the number of source domain images to match the quantity of target domain samples. By comparing results from MNIST to USPS and USPS to MNIST, it becomes clear that FDDS maintains its efficacy even when the source and target domains are interchanged. The adaptation learning data augmentation in depth domain, which is based on feature disentangling, brings the classifier closer to target-only model training. Additionally, the attention mechanism incorporated into the network aids in generating images during training and helps to differentiate features into content and style components. The results reveal that the model effectively separates the representation of content and style while preserving key features of each domain, thereby achieving outstanding results in digital classification that surpass those of most competing methods. Notably, in the MNIST to USPS task, FDDS's unsupervised digit recognition accuracy was 0.3% higher than that of supervised training using only the target domain.

### 4.2. Semantic Segmentation Task

4.2.1. Dataset

Compared with digit classification, semantic segmentation tasks are more complex. In order to compare with baselines and explore FDDS's domain adaptation performance in complex scenes, we used GTA5 [45] to Cityscapes [46] to complete the semantic segmentation task. The GTA5 and Cityscapes are 19 classes of classic public datasets, which have high-quality labeled images at the pixel level. Many methods (such as LC [14] and DRANet [15]) use GTA5 and Cityscapes as source and target domain, respectively, for unsupervised domain adaptation tasks. Compared with some methods, such as [47,48] which completed 13 classes of SYNTHIA [49] to the Cityscapes [46] task, our 19 classes task was more complex and better reflected the performance of domain adaptation.

The GTA5 [45] dataset is a computer-generated dataset of driving scenes, while the Cityscapes [46] dataset comprises genuine driving scenes captured in real-world environments. The GTA5 dataset contains 24,966 labeled RGB image samples sized at $1914 \times 1052$ pixels, with each image depicting an object or entity that commonly appears within 19 distinct classes. The Cityscapes dataset consists of 5000 labeled RGB images sized at $2040 \times 1016$ pixels, gathered from 50 different cities. Our next experiment used these two road scene datasets to explore the domain shift from synthetic images to real images.

4.2.2. Baselines and Implementation Details

- Source Only: The classifier trained in the source domain is directly used in the target domain;
- FCNs [50]: A classical pixel-level method for semantic segmentation using full convolution networks;
- CyCADA [13]: Combine features from features and pixels, and introduce cyclic loss into domain adaptive learning;
- SIBAN [51]: Classify by extracting shared features;
- LC [14]: It is equally important to put forward the lightweight calibrator component and start to pay attention to the performance of the source domain;
- DRANet [15]: Decouple features into style and content features, and propose nonlinear decoupling;
- Target Only: Training directly in the target domain and testing in the target domain is equivalent to supervised learning.

In the task of semantic segmentation, the initial network of FCN uses VVG-16-FCNS, and the rest of the methods use DRN-26 [52]. The training set of the source domain was used to train the network under supervision, and the task loss was used to make it a classifier of the source domain. In order to evaluate the above model, we used the source code provided by the author and some experimental data provided by SIBAN [51]. Our model, FDDS, was implemented in Pytorch. All the models in this paper were trained on an NVIDIA GTX 2080 GPU using CUDA11.7.

During the source domain classifier's training, we set the learning rate to $1 \times 10^{-3}$, SGD momentum to 0.9, and the batch size to 8. Under these conditions, 120 iterations were trained. Because semantic segmentation requires higher-quality images, we preprocessed the images in the dataset, adjusted the image size to $1024 \times 1024$ pixels, and used $400 * 400$ random pixel blocks for training. Due to memory limitations, we could only train one group of images at a time (a source domain image and a target domain image, the size of which is the same as the input size of domain shifter, which is $572 \times 572$). During the process of domain shifting, due to the large scale of the dataset and the complex task of semantic segmentation, we only trained 20 epochs in our experiment, and the running time was 8–10 h.

4.2.3. Experimental Process

In order to demonstrate the practicality of our model in complex real-world scenarios, we employed the GTA5 and Cityscapes datasets, which contain driving-scene images with dense annotations, representing a significantly more challenging task than the previous

digit classification task. Training our model on 24,966 images from the GTA5 dataset and 2975 images from the Cityscapes, we employed 19 common classes to facilitate adaptation from synthetic to real-world settings. Figure 4 depicts the mutual transformation between the source and target domains.

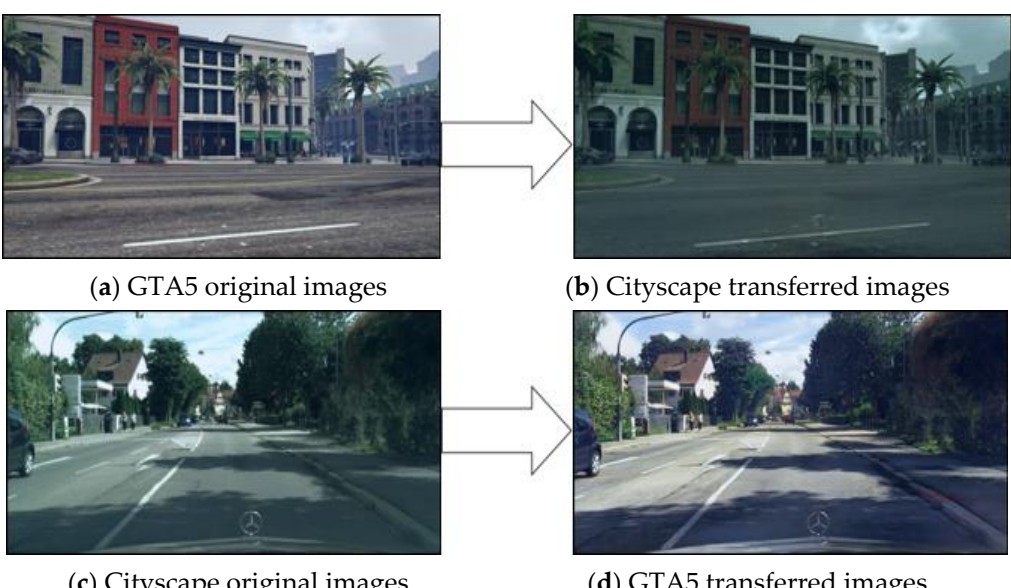

(**a**) GTA5 original images                    (**b**) Cityscape transferred images

(**c**) Cityscape original images                    (**d**) GTA5 transferred images

**Figure 4.** Image transfer between source and target domains.

To evaluate the semantic segmentation performance, we used three metrics: mean intersection-over-union (mIoU), frequency weighted intersection-over-union (fwIoU), and pixel accuracy (PixelAcc). We used the DRN-26 model [52] in CyCADA [13] as our source classifier, which was trained in stylized GTA5 images. To achieve adaptation from synthetic to real-world settings, we trained the DRN-26 model on 19 common classes.

In the task of GTA5 to Cityscape, we identified 19 classes, and compared FDDS with other competitive methods. Among them, FCN [50] is a classical semantic segmentation method, SIBAN [51] is a classification method that extracts shared features, and the linear/non-linear disentangling method was adopted from CyCADA [13], LC [14] and DRANet [15], which was the closest to our method. Numerical results reflecting the accuracy between FDDS and the competing methods under identical conditions are presented in Table 2.

Table 2 shows that our proposed method achieves superior results across all three main semantic segmentation metrics: mIoU, fwIoU and pixel accuracy PixelAcc. Among them, the pixel accuracy improved by 0.8%. There were 19 classes in the dataset, FDDS achieved optimal performance in 11 classes and optimal or suboptimal performance in 16 classes. Specifically, the accuracy of FDDS for the terrain label was improved by 1.1%, and that of the bicycle label was improved by 2.7%. In the case of single source domains and single target domains, our method performed well in the classes with obvious features (such as roads and buildings). This is because we used non-linear adaptation disentangling for feature comparison, which effectively reduces the differences between domains and improves the recognition accuracy of these classes. In contrast, although the domain shifter increased the number of samples compared with other methods, we also found the limitations of the FDDS method, which is that for classes with less training data or relatively changeable appearances, such as poles and trains, the classification performance lacks advantages.

**Table 2.** Accuracy comparison of FDDS to state-of-the-art methods for the semantic segmentation task on the road scene (%).

| | Road | Sidewalk | Building | Wall | Fence | Pole | Traffic Light | Traffic Sign | Vegetation | Terrain | Sky |
|---|---|---|---|---|---|---|---|---|---|---|---|
| Source Only | 42.7 | 26.3 | 51.7 | 5.5 | 6.8 | 13.8 | 23.6 | 6.9 | 75.5 | 11.5 | 36.8 |
| FCNs (2015) [50] | 70.5 | 32.3 | 62.2 | 14.8 | 5.4 | 10.8 | 14.3 | 2.7 | 79.3 | 21.2 | 64.6 |
| CyCADA (2018) [13] | 79.1 | 33.1 | 77.9 | 23.4 | 17.3 | 32.1 | **33.3** | **31.8** | 81.5 | 26.7 | 69.0 |
| SIBAN (2019) [51] | 83.4 | 13.1 | 77.8 | 20.3 | 17.6 | 24.5 | 22.8 | 9.7 | 81.4 | 29.5 | 77.3 |
| LC (2020) [14] | 83.5 | 35.2 | 79.9 | **24.6** | 16.2 | **32.8** | 33.1 | **31.8** | 81.7 | 29.2 | 66.3 |
| DRANet (2021) [15] | 83.5 | 33.7 | 80.7 | 22.7 | 19.2 | 25.2 | 28.6 | 25.8 | 84.1 | 32.8 | **84.4** |
| **Ours** | **84.1** | **35.7** | **80.9** | 23.5 | **20.7** | 26.7 | 29.0 | 27.5 | **84.5** | 33.9 | 79.6 |
| Target Only | 97.3 | 79.8 | 88.6 | 32.5 | 48.2 | 56.3 | 63.6 | 73.3 | 89.0 | 58.9 | 93.0 |
| | Person | Rider | Car | Truck | Bus | Train | Motorbike | Bicycle | mIoU | fwIoU | Pixel Acc. |
| Source Only | 49.3 | 0.9 | 46.7 | 3.4 | 5.0 | 0.0 | 5.0 | 1.4 | 21.7 | 47.4 | 62.5 |
| FCNs (2015) [50] | 44.1 | 4.3 | 70.3 | 8.0 | 7.2 | 0.0 | 3.6 | 0.1 | 27.1 | - | - |
| CyCADA (2018) [13] | 62.8 | 14.7 | 74.5 | 20.9 | 25.6 | 6.9 | 18.8 | 20.4 | 39.5 | 72.4 | 82.3 |
| SIBAN (2019) [51] | 42.7 | 10.8 | 75.8 | **21.8** | 18.9 | 5.7 | 14.1 | 2.1 | 34.2 | - | - |
| LC (2020) [14] | **63.0** | 14.3 | **81.8** | 21.0 | 26.5 | 8.5 | 16.7 | 24.0 | 40.5 | 75.1 | 84.0 |
| DRANet (2021) [15] | 53.3 | 13.6 | 75.7 | 21.7 | 30.6 | **15.8** | 20.3 | 19.5 | 40.6 | 75.6 | 84.9 |
| **Ours** | 52.9 | **15.3** | 75.8 | **21.8** | 31.3 | 9.7 | **20.9** | **26.7** | **41.1** | **76.2** | **85.7** |
| Target only | 78.2 | 55.2 | 92.2 | 45.0 | 67.3 | 39.6 | 49.9 | 73.6 | 67.4 | 89.6 | 94.3 |

The optimal performance is bold, and the suboptimal performance is underlined.

## 5. Discussion

### 5.1. Feature and Pixel-Level Domain Adaptation

Initially, we conducted an evaluation of transfer in both pixel and feature spaces. Our empirical investigations led us to the conclusion that when migrating USPS and MNIST datasets—two domains exhibiting a relatively small range of transfer learning in pixel space— utilizing images translated by CycleGAN proves highly effective. Indeed, this approach outperforms prior standard domain adaptation methods in terms of both performance and accuracy, and it is comparable to state-of-the-art domain adaptation approaches. Under these circumstances, pixel-level domain adaptation has proven highly advantageous. Conversely, when migrating from the source domain SVHN to the target domain MNIST, we found that feature-level domain adaptation significantly outperformed pixel-level domain adaptation. Consequently, it becomes beneficial to combine the two approaches, leveraging their respective strengths to produce a novel model with high performance across diverse domains.

### 5.2. Feature Disentangling Method

We propose a new feature separator that is non-linear and distinct from previous linear disentangling approaches. Our separator leverages the domain normalization factor to achieve separation. To demonstrate the effectiveness of our approach, we conducted various combination experiments in the framework, controlling variables and assessing network structure performance and classification results for domain adaptation tasks. The experimental results reveal showed that combining non-linear feature disentangling with the normalization factor yielded superior results than other experiments.

In the process of deep learning, normalization and standardization allow the data to better respond to the activation function and improve the expressiveness of the data. In our method, we set the normalization factor to 255, and map the RGB image [0, 255] to the [0, 1] interval during image preprocessing, so as to complete the normalization. Then, the mean and standard deviations of the RGB images were used to complete the standardization.

After this processing, the average value of the sample is 0 and the standard deviation is 1, which makes the model converge more easily.

In the domain adaptation task involving MNIST and MNIST-M datasets, since both datasets comprise identical content representations, all models exhibited reasonable performance even in the absence of non-linear and normalization factors. It is worth mentioning that MNIST-M denotes a variant of MNIST utilized for unsupervised domain adaptation, wherein background images are replaced, yet each MNIST number is preserved. Conversely, for MNIST and USPS adaptation, there is a significant divergence in content representation. In this case, models without these two components exhibited inadequate classification performance on one side, indicating that the model can only accommodate orientation domain adaptation (e.g., MNIST to USPS or USPS to MNIST) in the same manner as existing approaches. The numerical results are presented in Table 3.

**Table 3.** The effect of non-linear disentangling and normalization of our method (%).

| Non-Linear Disentangling | Normalization | USPS to MNIST | MNIST to USPS | SVHN to MNIST |
|:---:|:---:|:---:|:---:|:---:|
| | | 86.2 | 12.7 | 70.4 |
| | √ | 90.5 | 91.6 | 83.5 |
| √ | | 91.1 | 97.3 | 90.6 |
| √ | √ | 97.6 | 98.1 | 96.9 |

As shown in Table 3, our model introduced non-linear disentangling and normalization, which outperformed all the experimental conditions in three tasks. In the three tasks, nonlinear disentangling and normalization both improved the accuracy of domain adaptation. Our experiments indicate that non-linear mapping improves feature disentangling, leading to a significant enhancement in performance. As mentioned in [15], the non-linear mapping of features provides an advantage for clear separation and representation to a greater extent. Moreover, our findings showed that the normalization factor further enhances domain adaptation performance beyond the original experimental setup. Thus, we conclude that both factors play a crucial role in feature disentangling and unsupervised adaptation.

*5.3. Domain Shifter Component*

As previously discussed, one of the primary constraints of existing domain adaptation approaches is the inability of the same model to serve both the source and target domains. Typically, when confronted with a new target domain, most current domain adaptation methods require fine-tuning of the deployed model parameters [53,54]. However, the model running on the server operates within a specific environment and modifying parameters is not always feasible. Undoubtedly, adapting the running model to the new domain is a time-consuming and costly process unsuitable for time-sensitive applications. In comparison, our proposed method achieves adaptability without updating the running model but merely by integrating a domain shifter, offering greater flexibility when facing novel fields.

While some existing methods enhance target domain performance, they typically do so at the expense of source domain performance. In contrast, our approach maintains strong performance across both domains. To demonstrate the efficacy of our method, we assessed its performance against that of ADDA [9], CyCADA [13], and LC [14] before and after domain adaptation in the SVHN to MNIST task, as illustrated in Table 4.

**Table 4.** Comparison of some methods in SVHN to MNIST tasks (%).

| Method | Source Acc. (Before Adapt) | Source Acc. (After Adapt) | TargetAcc. (After Adapt) |
|---|---|---|---|
| ADDA (2017) [9] | 90.5 | 67.1 | 80.1 |
| CyCADA (2018) [13] | 92.3 | 31.4 | 90.4 |
| LC (2020) [14] | 93.9 | 90.8 | 97.1 |
| Ours | 94.1 | 92.6 | 96.9 |

Table 4 shows that ADDA [9] and CyCADA [13] fail to account for the source domain performance following domain adaptation. In addition, LC [14] utilizes a data calibrator to significantly enhance the source domain performance post-adaptation. In contrast, FDDS achieved nearly equivalent source domain performance post-adaptation relative to its performance prior to adaptation. Thus, FDDS offers clear advantages over competing methods.

In Section 3.1, we noted that domain adaptation methods face limitations in their ability to switch flexibly between source and target domains in certain instances. For models providing services, switching between the two domains demands significant resources, which may not be sufficient in time-sensitive applications. However, our proposed method circumvents this issue as it does not necessitate modifications to the model being served. By incorporating a domain shifter, our approach boasts greater versatility in adapting to new fields. Notably, our model is capable of operating in both source and target domains, which is made possible by the inclusion of the domain shifter.

Furthermore, the incorporation of a domain shifter results in moderate overhead. As a component of the network, the introduction of domain shifter into any neural network will inevitably increase the overhead. By comparing the number of parameters between the original network and the domain shifter, we can roughly estimate the overhead of adding the domain shifter. We tested the number of parameters of the classifier and the domain shifter. In the case of digit classification, LeNet comprises 3.1 million parameters, whereas the domain shifter accounts for only 0.19 million parameters, representing a mere 6.12% of the model's total parameters [14]. The relationship between domain shifters and the number of parameters in the original network are shown in Table 5.

**Table 5.** The number of parameters comparison of domain shifter to original network on two tasks.

| | Original Network (ON) | Num of Param. in ON (M) | Num of Param. in DS (M) | Radio of DS to ON (%) |
|---|---|---|---|---|
| Digit Classification | LeNet | 3.1 | 0.19 | 6.12 |
| Semantic Segmentation | DRN-26 | 20.6 | 0.06 | 0.29 |

DS represents domain shift, M represents million.

Table 5 shows the radio of the number of parameters of domain shifter in the original network. Due to different networks, the number of parameters of domain shifter is also different, but from our experimental results, the overhead of the domain shifter is indeed moderate compared with today's large network model. We therefore conclude that, when compared to the larger model being served, the domain shifter, a lightweight component, bolsters the model's ability to identify the source domain without imposing significant overhead on the network.

## 6. Conclusions

In this paper, we proposed a method of the feature disentangling and domain shifting (FDDS) for domain adaptation. We adopted a lightweight domain shifter component which stored the relevant information of the source domain by adding perturbation to the target domain and generated an image close to the source domain. We added a dual-attention

mechanism from the spatial and channel levels in the domain shifter to fuse features. In the feature disentangling, we used learnable weights to nonlinearly decompose a single feature into two parts, namely content and style, and recombined the two parts in different domains to generate domain-transfer images. In the process of domain transfer, FDDS utilized synthetic images to generate realistic domain-transfer images, and achieved performance in various visual recognition tasks, such as image classification and semantic segmentation. The domain shifter with the attention mechanism demonstrated the excellent network performance in the target domain while preserving the classification ability of the source domain, which verified the effectiveness of the attention mechanism in feature extraction in the image-generation field. The results showed that FDDS not only performed well in the digit classification task, but also had higher accuracy than previous methods in the complex cityscapes task. As an unsupervised domain adaptation method, the performance of FDDS was close to that of supervised learning using the target domain. In addition, FDDS is currently suitable for domain adaptation in both single source and single target domains; the limitation is that in the case of less training data, there is no advantage in the transfer effect and classification performance. In the future, this framework can be extended to domain adaptation across three or more domains and can use data augmentation to generate more data to achieve multi-directional transfer from the source domain to multiple target domains. Our work has research significance for domain adaptation methods in complex scenes, as well as in potential applications such as autonomous driving technology in automobile industry tasks.

**Author Contributions:** Conceptualization, H.C.; methodology, H.C. and H.C.; software, H.C.; validation, H.C. and F.G.; writing—original draft preparation, H.C.; writing—review and editing, F.G. and Q.Z.; project administration, F.G. and Q.Z. All authors have read and agreed to the published version of the manuscript.

**Funding:** This research was funded by the Zhejiang Provincial Natural Science Foundation of China (ZJNSF), grant numbers LY20E050011.

**Informed Consent Statement:** Informed consent was obtained from all subjects involved in the study.

**Data Availability Statement:** Not applicable.

**Conflicts of Interest:** The authors declare no conflict of interest.

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
