# Peer review of "FDDS: Feature Disentangling and Domain Shifting for Domain Adaptation"

_mathematics, doi:10.3390/math11132995_

Round 1

Reviewer 1 Report

The paper tries to present a new method for feature disentangling and domain shifting (FDDS) for domain adaptation. In the abstract we read that the authors evaluated FDDS against state-of-the-art algorithms on 21 digital and road scene datasets the results showed that in the 19 classification tasks for road scenes, FDDS achieves superior performance over the competition in 11 categories.

However, the paper is written in a careless way. Lines 102 to 107 seem to be pasted here from another paper. Figure 2, page 4 has nothing to do with Figure 2 mentioned in the above mentioned lines of text.

There a a lot of typos, for example the title of subsection 2.3, line 366, etc.

The numbers in tables 1,2,3 has no explanation. They should be the accuracy for those methods, but without any indication it could be considered as any other performance metrics.

Being an original method the authors should present some implementation details and also the code which are entirely missing.

The paper needs moderate English language editing. More, there are a lot of typos that must be corrected.

Author Response

The paper tries to present a new method for feature disentangling and domain shifting (FDDS) for domain adaptation. In the abstract we read that the authors evaluated FDDS against state-of-the-art algorithms on 21 digital and road scene datasets the results showed that in the 19 classification tasks for road scenes, FDDS achieves superior performance over the competition in 11 categories.

A: First of all, the authors thank the reviewers and editors for giving us the opportunity to continue revising the paper. We would also like to thank the reviewer for the comments and suggestions, which helped to improve the original manuscript.

Point 1:

The paper is written in a careless way. Lines 102 to 107 seem to be pasted here from another paper. Page 4 has nothing to do with Figure 2 mentioned in the above mentioned lines of text.

A1: We have removed the irrelevant sentences (Just used as a format template but forgot to delete it).

A2: We have rearranged the layout of the main text and figure to make them closer together.

A3: We have reconstructed Figure 2 and added a new paragraph to explain its contents.

Point 2:

The numbers in tables 1,2,3 has no explanation. They should be the accuracy for those methods, but without any indication it could be considered as any other performance metrics.

A1: The captions of all tables (Table 1 to Table 5) have been renamed to make them self-explanatory.

A2: We have added analysis and explanation of the table contents in the main text, which were addressed in Section 4.1, Section 4.2, Section 5.2, and Section 5.3, respectively.

Point 3:

Being an original method the authors should present some implementation details and also the code which are entirely missing.

A1: We have revised the Section 4 and added Section 4.1.2 and Section 4.2.2 to supplement the implementation details, which include the comparisons of the baseline methods in digit classification and road scene tasks, as well as the software of proposed model, running time, batch size and hyper-parameter setting.

A2: In Section 5.2, we have explained normalization progress to describe the selection of normalization factor and the function of normalization.

Point 4:

The paper needs moderate English language editing. More, there are a lot of typos that must be corrected (There are lots of typos, for example the title of subsection 2.3, line 366, etc.).

A: We have carefully checked and corrected the typos and grammar issues in the entire text, as best as we can.

Reviewer 2 Report

1.Deeplearning methods need to be elaboarted. 

2. Why only GTA5 & Cityscapes dataset were chosen?

3.  Results can be improved.

Paper requires proof reading. Also, check the punctuation

Reviewer 3 Report

The paper is technically sound and presents an interesting framework to support the domain adaptation strategies.

The literature review is shallow; the authors must include a Section focused on related works.

The resolution of Figure 2 must be substantially increased, and it is necessary to verify the aspect ratio. The term "experimental results" should be changed to "numerical results".

The paper requires deep proofreading. All equations must be double-checked all symbols must be defined. For instance,  the mathematical expectation symbol E{}  si not properly declared.

Table 1 uses some references; activating the corresponding hyperlink to jump directly to the bibliography would be nice.

Table 2 is not correctly formatted because it is combined with the line numeration. 

The caption on figures should be more self-descriptive, explaining the image contents. Besides, the font used in images must be similar in size to the current text.

The discussion should be included as a subsection into  the section devoted to presenting the results.

Many bibliographical references need to be updated; please use recent literature from the last five years from reputed journals.

Deep proofreading is required to improve style and description accuracy.

Author Response

The paper is technically sound and presents an interesting framework to support the domain adaptation strategies.

A: Thanks very much for the review’s positive comments.

Point 1:

The literature review is shallow; the authors must include a Section focused on related works.

A: According to the reviewer’s suggestion, we have added "2. Related Work". In this section, we have reviewed the literature survey related to our work, including “2.1 Deep Domain Adaptation”, “2.2 Attention Mechanism in Image Generation”, and “2.3 Feature Disentangling”.

Point 2:

The resolution of Figure 2 must be substantially increased, and it is necessary to verify the aspect ratio. The term "experimental results" should be changed to "numerical results".

A1: We have redrawn Figure 2 to increase the resolution and verify the aspect ratio.

A2: We have checked and corrected the term "experimental results" to "numerical results".

Point 3:

The paper requires deep proofreading. All equations must be double-checked all symbols must be defined. For instance, the mathematical expectation symbol E is not properly declared.

A1: We have carefully proofread the manuscript and checked the equations to ensure that all variables and symbols were defined when they first appeared.

A2: In Section 3.2, in order to avoid confusion, we have redefined the symbol of mathematical expectation (Lines 480-481 after Eq, 4).

Point 4:

Table 1 uses some references; activating the corresponding hyperlink to jump directly to the bibliography would be nice.

A: The function of hyperlink can jump directly to the bibliography and can provide readers with a more convenient reading experience. In fact, the Word version of our manuscript used Endnote to manage literatures, which were embedded with hyperlinks.

Point 5:

Table 2 is not correctly formatted because it is combined with the line numeration.

A: We have redrawn and rearranged Table 2 in the revision.

Point 6:

The caption on figures should be more self-descriptive, explaining the image contents. Besides, the font used in images must be similar in size to the current text.

A: Thanks for the reviewer’s suggestion. Firstly, we have checked and revised the caption on figures to make it self-explanatory. In addition, we have adjusted the font and size in the image to match the current text.

Point 7:

The discussion should be included as a subsection into the section devoted to presenting the results.

A: According to the reviewer's suggestion, we have divided the "results and discussion" into two sections, namely, Sections "4 Results" and "5 Discussions". The related contents have also been rearranged.

Point 8:

Many bibliographical references need to be updated; please use recent literature from the last five years from reputed journals.

A: We have updated the relevant literatures to summarize and cover recent research findings. In addition, we have added a literature review section on related work in Section 2.

Reviewer 4 Report

The article investigates the feature disentangling and domain shifting for domain adaptation. After the introduction part, the authors present the model description, domain shifter network, feature distangling module. In the results and discussion part the dataset, the experimental process, the semantic segmentation, Ablation Study are discussed. The last section is the conclusions part. I have the folowing comments:

·        The introduction section should be extended with more related papers

·        From the results and discussion section the presentation of the software implementation is missing.

·        The running time of the methods are also not presented

·        Table 2: Why does it contain only so many methods?

·        Figure 4: instead of a figure a table would be better

·        The practical implication of the study should be more highlighted

·        Conclusions section: too short and future work is also missing

Author Response

Point 1:

The introduction section should be extended with more related papers. I do not recommend specific articles. But I recommend using the following keywords for searching related papers: Feature Disentangling, Domain Shifting, Domain Adaptation, attention mechanism. A comparison of the articles can also be illustrated with a table or figure.

A: According to the reviewer’s suggestion, we have extended introduction section and added "2. Related Work". In this section, we have reviewed the literature survey related to our work, including “2.1 Deep Domain Adaptation”, “2.2 Attention Mechanism in Image Generation”, and “2.3 Feature Disentangling”.

Point 2:

From the results and discussion section the presentation of the software implementation is missing. Which software was used and why? The algorithm is own implementation? If yes, the following question can be also considered: software structure (e.g. UML diagram). The running time of the methods are also not presented.

A1: The algorithm is own implementation. The network structure of the algorithm/software has been shown in Figure 1.

A2: We have revised the Section 4 and added Section 4.1.2 and Section 4.2.2 to supplement the software implementation in detail, which include the software of proposed model, running time, batch size and hyper-parameter setting, etc.

Point 3:

Why were the methods included in Table 1 examined? The brief presentation of these methods is also missing. (Source Only, DANN, DSN, ADDA, CyCADA, LC, DRANet, GTA, CDA, Ours)

A1: In the added Section 4.1.2, we have briefly described these methods and explained the reasons for the comparison. To demonstrate the competitiveness of the proposed method, these comparisons include classic, closely related to our work, and recently available methods.

A2: The brief presentation of these methods have been added in Section 4.1.3. Compared FDDS with these previous methods, in which Source Only and Target Only were used as the control group, which respectively represented the results of training only using source/target domain; DANN, DSN, ADDA and GTA used traditional feature separation methods to separate features into private and shared parts, CyCADA and DRANet separated features into style and content, and LC and CDA used lightweight components and confrontation generation networks to realize domain adaptation.

Point 4:

Table 2: Why does it contain only so many methods? (Source Only, CyCADA, LC, DRANet, Ours)? Why only GTA5 and Cityscapes datasets are examined?

A1: Similar to Point 3, in the added Section 4.2.2, we have briefly described these methods and explained the reasons for the dataset selection.

A2: In order to make the comparison more representative, we have added two methods in Table 2, namely FCNs and SIBAN, respectively.

Point 5:

Figure 4: instead of a figure a table would be better, because with a table the numerical values can be better presented.

A: According to the reviewer’s suggestion, The Figure 4 (in original version) has been converted to Table 4 in Section 5.3.

Point 6:

The practical implication of the study should be more highlighted. What industrial tasks, in which area can this research be used for? Conclusions section: too short and future work is also missing. A more detailed summary of the achieved results is necessary.

A: We have rewritten the conclusions. In this section, the proposed method and achieved results were summarized in detail. In addition, the limitation was mentioned and the further work in the future were extended. Finally, the research significance for the learning method and practical implication for industrial tasks was highlighted.

The authors would like to express their greatest appreciation to the Reviewers for their careful review of the paper and, in particular, for their useful and constructive comments and suggestions for the revised version of the paper.

Round 2

Reviewer 1 Report

The authors made some improvements. i consider that the paper can be published in the present form.

English is acceptable.

Reviewer 3 Report

The authors have addressed all my previous concerns. The equations have been corrected and image quality improved. Besides, issues related to the tables and captions were solved satisfactorily. 

The manuscript has been substantially improved, reducing significantly the previous typographical errors.